



# Split-explicit external mode solver in finite volume sea ice ocean model FESOM2

Tridib Banerjee[1,2], Patrick Scholz[1], Sergey Danilov[1,2], Knut Klingbeil[3], and Dimitry Sidorenko[1]

[1]Alfred Wegener Institute, Helmholtz Centre for Polar and Marine Research, Bremerhaven, Germany
[2]Constructor University, Bremen, Germany
[3]Leibniz-Institute for Baltic Sea Research, Rostock, Germany

**Correspondence:** tridib.banerjee@awi.de

**Abstract.** A novel split-explicit (SE) external mode solver for the Finite volumE Sea ice–Ocean Model (FESOM2) and its sub-versions (example 2.5) is presented. It is compared with the semi-implicit (SI) solver currently used in FESOM2. The split-explicit solver utilizes a dissipative asynchronous (forward–backward) time-stepping scheme. Its implementation with Arbitrary Lagrangian-Eulerian vertical coordinates like Z-star ($Z^*$) and Z-tilde ($\tilde{Z}$) is explored. The comparisons are performed through multiple test cases involving idealized and realistic global simulations. The SE solver demonstrates lower phase errors and dissipation, but maintain a simulated mean ocean state very similar to the SI solver. The SE solver is also shown to possess better run-time performance and parallel scalability across all tested workloads.

## 1 Introduction

All version 2 iterations of the Finite volumE Sea ice-Ocean Model (FESOM2; Danilov et al. (2017)) as its predecessor FE-SOM1.4, relies on an implicit algorithm for solution to the external mode. Its computational algorithm maintains elementary options of the Arbitrary Lagrangian Eulerian (ALE) vertical coordinate, such as $z^*$ or non-linear free surface, but needs modifications to incorporate more general options, beginning from $\tilde{z}$, where information on horizontal divergence in scalar cells is used when taking decision about layer thicknesses on a new time level in the internal (baroclinic) mode. This work aims to present the modified time-stepping algorithm and its extension through a split-explicit option for solution to the external mode. Many modern ocean circulation models rely on the split-explicit method to solve for its external mode. The primary motivation behind such a choice is expectation of better parallel scalability in massively parallel applications. Indeed, as is well known, the need for global communications to calculate certain global dot products, in most iterative solvers is a factor that potentially slows down the overall performance (see e.g., Huang et al. (2016), Koldunov et al. (2019)). Although there are solutions minimizing the number of global communications per iteration (see, e. g., Cools and Vanroose (2017)), as well as solutions where global communications are avoided (e.g. Huang et al. (2016)), split-explicit methods are an obvious alternative. It is followed by GFDL Global Ocean and Sea Ice Model OM4 whose ocean component uses version 6 of the Modular Ocean Model MOM (Adcroft et al. (2019)), Nucleus for European Modelling of the Ocean NEMO (Madec et al. (2019)), Regional Oceanic Modeling System ROMS (Shchepetkin and McWilliams (2005)) and Model for Prediction Across Scales Ocean MPAS-O (Ringler et al. (2013)), to mention just some widely used cases.





A careful analysis in Shchepetkin and McWilliams (2005) discusses many details of the numerical implementation for a split-explicit external mode algorithm, and proposes the AB3-AM4 (Adams–Bashforth and Adams–Moulton) method which is at present followed by several models (ROMS Shchepetkin and McWilliams (2005), CROCO Jullien et al. (2022), FESOM-C Androsov et al. (2019) etc). However, recent analysis in Demange et al. (2019) suggests a simpler choice of dissipative forward–backward time-stepping. The built-in dissipation in this case allows one to avoid filtering of the external mode solution (see

Shchepetkin and McWilliams (2005)). The simplest commonly used filter requires that an external mode be stepped across two baroclinic time steps to ensure temporal centering. This doubles the computational cost of the external mode solution. In the forward-backward dissipative method by Demange et al. (2019) an external mode is stepped precisely across one baroclinic time step and not beyond. This method was ultimately found to be the best choice to build a split-explicit scheme around for our purposes. Demange et al. (2019) also showed how dissipation can be added to the AB3-AM4 method of Shchepetkin and

McWilliams (2005). We thus also explore dissipative AB3-AM4 for dissipation and phase errors.

The rest of the paper is thus structured as follows. We begin with providing a breakdown for individual steps of Split-Explicit schemes as adopted in FESOM2 (section 2). It is then followed by comparing the individual temporal interpolations that are characteristic of these time-stepping schemes (section 3). We then perform various experiments comparing the new solver against the existing one over different test cases (sections 4 and 5). Finally, we summarize the results and argue for the new

proposed scheme and its solver being a great choice for FESOM2 moving forward (section 7).

## 2 Split-explicit asynchronous time-stepping

This section provides a detailed description of the proposed asynchronous time-stepping scheme for FESOM2 that incorporates a split-explicit barotropic solver. FESOM in its standard version relies on a semi-implicit barotropic solver which already uses an asynchronous time-stepping (Danilov et al., 2017). The asynchronous time-stepping is a variant of forward-backward

time-stepping which is formulated by considering scalar and horizontal velocities as being displaced by half a time-step $\tau/2$. The asynchronous time stepping is taken as the simplest option. Other time stepping options for the baroclinic part such as 3rd order Runge-Kutta method are under consideration for future versions.

### 2.1 Momentum equation

The standard set of equations under the Boussinesq and standard approximations is solved. The equations are taken in a layer-integrated form, and the placement of the variables on the mesh is explained in Danilov et al. (2017). The layer-integrated momentum equation in the flux form is,

$$\partial_t \mathbf{U}_k + \nabla_h \cdot (\mathbf{U}\mathbf{u})_k + (w^t \mathbf{u}^t - w^b \mathbf{u}^b)_k + f\mathbf{e}_z \times \mathbf{U}_k + h(\nabla_h p + g\rho \nabla_h Z)_k/\rho_0 = (V_h \mathbf{U} + (\nu_v \partial_z \mathbf{u})^t - (\nu_v \partial_z \mathbf{u})^b)_k \tag{1}$$

with $\mathbf{U}_k = \mathbf{u}_k h_k$ the horizontal transports, $\mathbf{u}$ the horizontal velocity, $h_k$ the layer thickness $V_h$ the horizontal viscosity operator,

$\nu_v$ the vertical viscosity coefficient, $f$ the Coriolis parameter, $\mathbf{e}_z$ a unit vertical vector, and $\nabla_h = (\partial_x, \partial_y)$ with respect to a





constant model layer. Here $k$ is the layer index, starting from 1 in the surface layer and increasing downward to the available number of levels with maximum value $N_l$. We ignore the momentum source due to the added water $W$ at the surface. The term with the pressure gradient, $g\rho\nabla_h Z_k$, accounts for the fact that layers deviate from geopotential surfaces. The quantity $Z_k$ appearing in this term is the $z$-coordinate of the midplane of the layer with the thickness $h_k$. The equation for elevation is
60 written as,

$$\partial\eta + \nabla_h \cdot \overline{\mathbf{U}} = W \tag{2}$$

where $\overline{\mathbf{U}} = \sum_k \mathbf{U}$, and equations for layer thicknesses $h_k$ and tracers will be presented further. Equations further in this section are for a particular layer $k$ and the index $k$ will suppressed. In the implementation described here, the discrete scalar state variables (elevation $\eta$, temperature $T$, salinity $S$ and layer thicknesses $h$) are defined at full time steps denoted by the
65 upper index $n$, whereas 3D velocities $\mathbf{v} = (\mathbf{u}, w)$ and horizontal transports $\mathbf{U}$ are defined at half-integer time steps $(n+1/2, \ldots)$. Since thicknesses and horizontal velocities are not synchronous, layer transports $\mathbf{U}$ are chosen as prognostic variables. By using them we avoid the question on $h^{n+1/2}$ up to the moment of barotropic correction. Note that the flux form of the momentum advection is used in equation (1). Adjustments needed for other forms are straightforward and will not be discussed here.

First, we estimate the transport $\mathbf{U}^{n+1/2,*}$ assuming that $h^n, T^n, S^n, \eta^n, \mathbf{u}^{n-1/2}, \mathbf{U}^{n-1/2}$ and $w^{n-1/2}$ are known.

$$\mathbf{U}^{n+1/2,*} - \mathbf{U}^{n-1/2} = \tau[(\mathbf{R}_{\mathbf{U}}^A + \mathbf{R}_{\mathbf{U}}^C)^n + (\mathbf{R}_{\mathbf{U}}^P)^n + (\mathbf{R}_{\mathbf{U}}^{hV})^{n-1/2}] \tag{3}$$

The terms $\mathbf{R}_{\mathbf{U}}^i$ with $i = A, C, P, hV$ indicate advective, Coriolis, pressure gradient, and horizontal viscosity components estimated at time step $n$ for $i = A, C, P$ and $n - 1/2$ for the horizontal viscosity. The momentum advection term is,

$$\mathbf{R}_{\mathbf{U}}^A = -\nabla(\mathbf{u}\mathbf{U}) - w\mathbf{u}|_b^t$$

where $|_b^t$ implies that the difference between the top and bottom interfaces of layer $k$ is taken. The Coriolis term is,

$$\mathbf{R}_{\mathbf{U}}^C = -f\mathbf{e}_z \times \mathbf{U}$$

Fields entering these advection and Coriolis terms are known at $n-1/2$ and the second or third-order Adams-Bashforth method is used to get an estimate of $\mathbf{R}_{\mathbf{U}}^A$ and $\mathbf{R}_{\mathbf{U}}^C$ at $n$. For any quantity $f$, $f^{AB} = (3/2+\beta)f^n - (1/2+2\beta)f^{n-1} + \beta f^{n-2}$. For classical third-order interpolation (AB3), $\beta$ is $5/12$, and $\beta = 0$ gives the second-order result (AB2). In the pressure gradient force,

$$\mathbf{R}_{\mathbf{U}}^P = -h\nabla_z p/\rho_0 = -h\nabla_h p/\rho_0 - hg\nabla_h Z$$

where $\nabla_z$ means differencing at constant $z$. Since pressure and thicknesses are known at the time level $n$, no interpolation is needed. This is one of the advantages of asynchronous time-stepping. Depending on how much layer thicknesses are perturbed, other algorithms than written above can be applied to minimize pressure gradient errors. As a default, the approach by Shchepetkin and McWilliams (2003) is used in FESOM. The pressure contains contributions from $\eta$, density perturbations in the fluid column, as well as contributions from atmospheric and ice loading. The contribution from horizontal viscosity is either of harmonic or biharmonic type. For simplicity, we write it here as,

$$\mathbf{R}_{\mathbf{U}}^{hV} = \nabla_h \cdot (hA_h\nabla_h\mathbf{u})$$





The implicit contribution from vertical viscosity is added as,

$$\mathbf{U}^{n+1/2,**} - \mathbf{U}^{n+1/2,*} = \tau\mathbf{R}_{\mathbf{U}}^{vV} = \tau(A_v\partial_z\mathbf{u}^{n+1/2,**})|_b^t \tag{4}$$

The latter equation is rewritten for increments $\Delta\mathbf{u} = \mathbf{u}^{n+1/2,**} - \mathbf{u}^{n+1/2,*} = (\mathbf{U}^{n+1/2,**} - \mathbf{U}^{n+1/2,*})/h^*$ and solved for $\Delta\mathbf{u}$. At this stage of the scheme, a reliable estimate for $h^*$ at $n+1/2$ is not available but, since $A_v$ is a parameterization and this step is first-order in time, we use $h^* = h^n$. When $\Delta\mathbf{u}$ is obtained, we update $\tau\mathbf{R}_{\mathbf{U}}^{vV} = \Delta\mathbf{u}h^n$ and $\mathbf{U}^{n+1/2,**} = \mathbf{U}^{n+1/2,*} + \Delta\mathbf{u}h^n$. In preparation for the barotropic time step, the vertically integrated forcing from baroclinic dynamics is computed as,

$$\overline{\mathbf{R}}^n = \sum_k [(\mathbf{R}_{\mathbf{U}}^A)^n + (\tilde{\mathbf{R}}_{\mathbf{U}}^P)^n + (\mathbf{R}_{\mathbf{U}}^{hV})^{n-1/2} + (\mathbf{R}_{\mathbf{U}}^{vV})^{n+1/2}]_k \tag{5}$$

Here, $\tilde{R}_U^P$ represents the pressure gradient force excluding the contribution of $\eta$ as it will be accounted for explicitly in the barotropic equation. The Coriolis term is also omitted for the same reason. The vertically summed contribution from vertical viscosity is in reality the difference in surface stress and bottom stress. The bottom stress in FESOM is commonly computed as $C_d|\mathbf{u}_b^{n-1/2}|\mathbf{u}_b^{n-1/2}$, where $\mathbf{u}_b$ is the bottom velocity.

## 2.2 Barotropic time-stepping

Next is the barotropic step where $\eta$ and $\overline{\mathbf{U}} = \sum_k \mathbf{U}_k$ are estimated by solving,

$$\partial_t\overline{\mathbf{U}} + f\mathbf{e}_z \times \overline{\mathbf{U}} + gH\nabla\eta = \overline{\mathbf{R}}, \quad \partial_t\eta + \nabla_h\overline{\mathbf{U}} + W = 0 \tag{6}$$

Here $H = H_0 + \eta$, $W$ is the freshwater flux (positive out of ocean), and $\overline{\mathbf{R}}$ the is the forcing from the 3D part defined above. These equations are solved from time step $n$ to $n+1$ as detailed below. Note that the baroclinic forcing term is taken at time level $n$. Centering it at $n+1/2$ would have involved a lot of additional computations and is not implemented at present. This set of equations present a minimum model. It is sufficient for basins with simple geometry. In realistic applications it has been found that an additional viscous regularization term is needed to suppress oscillations in narrow straits with irregular coastline. In such cases we add,

$$\overline{\mathbf{R}}^{hV} = \nabla_h \cdot (H\overline{A}_h\nabla_h\overline{\mathbf{U}}/H) \tag{7}$$

to the right hand side of momentum equation (6), and subtract the initial value of this term from $\overline{\mathbf{R}}$ on each baroclinic time step. Here $\overline{A}_h$ is the viscosity coefficient tuned experimentally to ensure stability in narrow shallow regions. We express it as a combination of some background viscosity and a flow-dependent part which is proportional to the differences of barotropic velocity across cell edges. The mentioned subtraction serves to minimize the inconsistency created by adding the new term. Note that in coastal applications, one generally keeps bottom drag acting on the barotropic flow as well as barotropic momentum advection (Klingbeil et al., 2018). We treat them as slow processes here, but modifications might be needed for possible future applications.





As a default time-stepping for the barotropic part the forward-backward dissipative time-stepping by Demange et al. (2019)
is used. It is abbreviated as SE (for split-explicit) further.

$$\overline{\mathbf{U}}^{n+(m+1)/M} = \overline{\mathbf{U}}^{n+m/M} - (\tau/M)[(1/2)fe_z \times (\overline{\mathbf{U}}^{n+(m+1)/M} + \overline{\mathbf{U}}^{n+m/M}) - gH^{n+m/M}\nabla_h\eta^{n+m/M} - \overline{\mathbf{R}}^n]$$
$$\eta^{n+(m+1)/M} = \eta^{n+m/M} - (\tau/M)\nabla_h \cdot \left[(1+\theta)\overline{\mathbf{U}}^{n+(m+1)/M} - \theta\overline{\mathbf{U}}^{n+m/M}\right] \tag{8}$$

Here $M$ is the total number of barotropic substeps per the baroclinic step $\tau$, and $\theta$ controls dissipation. The value of $\theta = 0.14$ is
mentioned by Demange et al. (2019) as being sufficient. We also used another version that is based on the AB3-AM4 (Adams-
Bashforth – Adams-Moulton) approach of Shchepetkin and McWilliams (2005) with dissipative corrections as proposed in
Demange et al. (2019) (abbreviated as SESM further). The specific versions of AB3 and AM4 used are,

$$f^{AB3} = (3/2+\beta)f^m - (1/2+2\beta)f^{m-1} + \beta f^{m-2}$$
$$f^{AM4} = \delta f^{m+1} + (1-\delta-\gamma-\zeta)f^m + \gamma f^{m-1} + \zeta f^{m-2} \tag{9}$$

with appropriate values of $\beta, \delta, \gamma, \zeta$ discussed later. The time-stepping takes the form,

$$\eta^{n+(m+1)/M} - \eta^{n+m/M} = (\tau/M)[-\nabla_h \cdot \overline{\mathbf{U}}^{AB3} - W]$$
$$\overline{\mathbf{U}}^{n+(m+1)/M} - \overline{\mathbf{U}}^{n+m/M} = (\tau/M)[-fe_z \times \overline{\mathbf{U}}^{AB3} - gH^{AM4}\nabla_h\eta^{AM4} + \overline{\mathbf{R}}^n] \tag{10}$$

### 2.3 Reconcilation of barotropic and baroclinic mode

Note that the use of dissipative time-stepping in (8) or (10) allows one to abandon filtering of $\eta$ and $\overline{\mathbf{U}}$ at the end of the
barotropic step that would be needed if non-dissipative forward-backward ($\theta = 0$) or the original AB3-AM4 schemes were
applied instead (see Shchepetkin and McWilliams (2005)). The most elementary form of filtering involves integration to $n+$
2 with subsequent averaging to $n+1$, which would double the computational expenses for the barotropic solver. To be in
agreement with the traditional notation, we write $\langle\eta\rangle^{n+1} = \eta^{n+m/M}$ and $\langle\overline{\mathbf{U}}\rangle^{n+1} = \overline{\mathbf{U}}^{n+m/M}$ for $m = M$ (there would be a
difference if filtering were needed). By summing the elevation equations over $M$ substeps, one gets for the forward-backward
dissipative case (8),

$$\langle\eta\rangle^{n+1} - \langle\eta\rangle^n = -\tau\nabla_h \cdot \langle\langle\overline{\mathbf{U}}\rangle\rangle^{n+1/2} \tag{11}$$

where,

$$\langle\langle\overline{\mathbf{U}}\rangle\rangle^{n+1/2} = \frac{1}{M}\sum_{m=1}^{M}\overline{\mathbf{U}}^{n+m/M} + \frac{\theta}{M}\left(\langle\overline{\mathbf{U}}^{n+1}\rangle - \langle\overline{\mathbf{U}}^n\rangle\right) \tag{12}$$

While $\eta$ is consistently initialized with $\langle\eta\rangle^n$ for $m = 0$, there is no good answer for $\overline{\mathbf{U}}$. One can use the last available $\langle\overline{\mathbf{U}}\rangle^n$
but, because 3D and barotropic velocities are integrated using different methods, this may lead to divergences with time unless
some synchronization with 3D velocities is foreseen. We return to this topic below. On time level $n+1$ the total thickness





becomes $H^{n+1} = H^0 + \langle \eta \rangle^{n+1}$. The horizontal transport is finalized by making the vertically integrated transport equal to the value obtained from the barotropic solution.

$$\mathbf{U}_k^{n+1/2} = \mathbf{U}_k^{n+1/2,**} - \frac{h_k^{n+1/2}}{\sum_k h_k^{n+1/2}} \left( \sum_k \mathbf{U}_k^{n+1/2,**} - \langle\langle \overline{\mathbf{U}} \rangle\rangle^{n+1/2} \right) \tag{13}$$

### 2.4 Finalization of baroclinic mode

The estimate of the thickness at $n+1/2$ depends on the option of the ALE vertical coordinate and will be detailed further. In treating the scalar part we are relying on the V-ALE approach in the terminology of Griffies et al. (2020). It is assumed that there is some external procedure to predict $h_k^{n+1} = h_k^{target}$ constrained by the condition $\sum_k h_k^{n+1} = H^0 + \langle \eta \rangle^{n+1}$. In the

130 simplest case, this is the $z^*$ vertical coordinate with $h_k^{n+1} = h_k^0 (H/H^0)$. Here, as well as in other cases when the decision on $h^{target} = h^{n+1}$ does not depend on layer horizontal divergences, $h^{n+1/2}$ in (13) is half sum of $n$ and $n+1$ values. In more complicated cases, such as $\tilde{z}$ (Leclair and Madec (2011), Petersen et al. (2015), Megann et al. (2022)), the horizontal divergence in layers $\nabla \cdot \mathbf{U}_k^{n+1/2}$ is needed to predict $h_k^{n+1}$, and a reliable estimate of $h^{n+1/2}$ is not immediately available. Enforcing that $h_k^{n+1}$ is smooth and positive and also satisfies the barotropic constraint $\sum_k h_k^{n+1} = H^0 + \langle \eta \rangle^{n+1}$ could be a non-trivial task

and may require a special procedure (see Hallberg and Adcroft (2009) and Megann et al. (2022)) which simultaneously adjusts $\mathbf{U}_k^{n+1/2}$ and $h^{n+1}$. The description of the current implementation of $\tilde{z}$ in FESOM is presented in Appendix B. The potential presence of such complications is the reason why the decision on $h^{n+1}$ is delayed to the end and the discretization of momentum equation is performed in terms of $\mathbf{U}$. Once the new thickness is determined, the thickness equation,

$$h_k^{n+1} = h_k^n - \tau [\nabla_h \cdot \mathbf{U} + w|_b^t]_k - \tau W \delta_{k1} \tag{14}$$

is used to estimate the diasurface velocity $w$. Tracers are then advanced first taking into account advection and horizontal (isoneutral) diffusion before being trimmed by implicit vertical diffusion.

$$(h^{n+1}T^*)_k = (hT)_k^n - \tau[\nabla(UT) + (wT)|_b^t]_k - \tau W T_W \delta_{k1} + (\nabla(hK)\nabla T)_k^n$$
$$(h^{n+1}T^{n+1})_k = (h^{n+1}T^*)_k + \tau(K_v \partial_z T^{n+1})_k|_b^t \tag{15}$$

Here $T_W$ it the value of scalar $T$ in freshwater flux. In this procedure, if $T^n = \text{const}$, the second equation will return this constant in $T^*$. The two equations above could have being combined into a single one. We treat them separately to avoid loss

of some significant digits (and ensuing errors in constancy preservation). Before solving the last equation in (15), it is rewritten for the increment $\Delta T = T^{n+1} - T^*$.

While $\mathbf{U}_k^{n+1/2}$ trimmed as given by (13) ensures by virtue of the first equation in (15) that $\sum_k h_k^{n+1} = h^0 + \langle \eta \rangle^{n+1}$ as required for perfect volume conservation, its vertical sum $\sum_k \mathbf{U}_k^{n+1/2} = \langle\langle \overline{\mathbf{U}} \rangle\rangle^{n+1/2}$ deviates from the barotropic transport at time level $n+1/2$ (i.e for $m = M/2$). We tried to compensate for this difference, by saving $\overline{\mathbf{U}}^{n+m/M}$ for $m = M/2$ in

the barotropic step and re-trimming the 3D transports once tracers are advanced, but this has been found to be redundant in practice. The number of barotropic substeps $M$ depends on the quality of meshes with varying resolution. It can always be estimated based on mesh cell size and local depth. The presence of particularly small cells on deep water may limit the scheme





globally even if the rest of the mesh is regular. Such limitations are absent in the present version of FESOM that is based on an implicit barotropic solver. Appendix A summarizes the changes needed to extend it (Danilov et al. (2017)) to more general
ALE options.

## 3 Temporal Interpolations for Barotropic Solver

This section provides a detailed numerical analysis of the new external mode solver when using SE, or SESM time-stepping against the current Semi-Implicit solver (Danilov et al. (2017)) of FESOM2 which uses a first-order implicit time-stepping in global simulations. Although parts of these implementations are already known, we repeat them here for clarity and compari-
160 son. A simple prototype system relevant for this analysis is,

$$\partial_t \tilde{u} = -c_p \partial_x \tilde{\eta}$$
$$\partial_t \tilde{\eta} = -c_p \partial_x \tilde{u} \tag{16}$$

where $c_p = (gH_0)^{1/2}$ is the phase velocity, $\tilde{u}$ the dimensionless vertically averaged velocity $(\bar{u}/c_p)$, and $\tilde{\eta}$ the dimensionless surface elevation $(\eta/H)$. It will be assumed that $\tilde{\eta}, \tilde{u} \sim e^{ikx}$, where $k$ is the wave number.

### 3.1 Characteristic Matrix Forms

### 3.1.1 Semi-implicit method

We begin with the Semi-Implicit method used in current FESOM2 Danilov et al. (2017) which was adapted from FESOM 1.4 Wang et al. (2014) and can be written as,

$$\tilde{\eta}^{n+1} = \tilde{\eta}^n - ic \left[ \alpha \tilde{u}^{n+1} + (1-\alpha) \tilde{u}^n \right]$$
$$\tilde{u}^{n+1} = \tilde{u}^n - ic \left[ \theta \tilde{\eta}^{n+1} + (1-\theta) \tilde{\eta}^n \right] \tag{17}$$

Here, $1/2 \leq \theta, \alpha \leq 1$ are control parameters and $c = c_p k \tau$ is the Courant number. The characteristic matrix form for this
scheme is,

$$\begin{Bmatrix} \tilde{\eta}^{n+1} \\ \tilde{u}^{n+1} \end{Bmatrix} = \frac{1}{\alpha\theta c^2 + 1} \begin{bmatrix} \alpha\theta c^2 - \alpha c^2 + 1 & -ic \\ -ic & \alpha\theta c^2 - \theta c^2 + 1 \end{bmatrix} \begin{Bmatrix} \tilde{\eta}^n \\ \tilde{u}^n \end{Bmatrix} \tag{18}$$

### 3.1.2 Explicit method of Shchepetkin and McWilliams (2005)

For the explicit method of Shchepetkin and McWilliams (2005) based on an advanced forward-backward method combining AB3 and AM4 steps, it can be expressed as,

$$\tilde{\eta}^{n+1} = \tilde{\eta}^n - ic \left[ (3/2 + \beta) \tilde{u}^n - (1/2 + 2\beta) \tilde{u}^{n-1} + \beta \tilde{u}^{n-2} \right]$$
$$\tilde{u}^{n+1} = \tilde{u}^n - ic \left[ \delta \tilde{\eta}^{n+1} + (1 - \delta - \gamma - \zeta) \tilde{\eta}^n + \gamma \tilde{\eta}^{n-1} + \zeta \tilde{\eta}^{n-2} \right] \tag{19}$$



Here too, $\beta, \delta, \gamma, \zeta$ are the control parameters. Its characteristic matrix form is then,

$$
\begin{Bmatrix} \tilde{\eta}^{n+1} \\ \tilde{\eta}^{n} \\ \tilde{\eta}^{n-1} \\ \tilde{u}^{n+1} \\ \tilde{u}^{n} \\ \tilde{u}^{n-1} \end{Bmatrix} = \begin{bmatrix} 1 & 0 & 0 & -ic(3/2+\beta) & ic(1/2+2\beta) & -ic\beta \\ 1 & 0 & 0 & 0 & 0 & 0 \\ 0 & 1 & 0 & 0 & 0 & 0 \\ -ic(1-\gamma-\zeta) & -ic\gamma & -ic\zeta & 1-c^2\delta(3/2+\beta) & c^2\delta(1/2+2\beta) & -c^2\delta\beta \\ 0 & 0 & 0 & 1 & 0 & 0 \\ 0 & 0 & 0 & 0 & 1 & 0 \end{bmatrix} \begin{Bmatrix} \tilde{\eta}^{n} \\ \tilde{\eta}^{n-1} \\ \tilde{\eta}^{n-2} \\ \tilde{u}^{n} \\ \tilde{u}^{n-1} \\ \tilde{u}^{n-2} \end{Bmatrix} \tag{20}
$$

### 3.1.3 Split-explicit method by Demange et al. (2019)

Finally, the SE method by Demange et al. (2019) can be expressed as,

$$
\tilde{\eta}^{n+1} = \tilde{\eta}^{n} - ic\left[(1+\theta)\tilde{u}^{n+1} - \theta\tilde{u}^{n}\right]
$$

$$
\tilde{u}^{n+1} = \tilde{u}^{n} - ic\tilde{\eta}^{n} \tag{21}
$$

with $\theta$ being the control parameter. Its characteristic matrix form is,

$$
\begin{Bmatrix} \tilde{\eta}^{n+1} \\ \tilde{u}^{n+1} \end{Bmatrix} = \begin{bmatrix} 1-c^2(1+\theta) & -ic \\ -ic & 1 \end{bmatrix} \begin{Bmatrix} \tilde{\eta}^{n} \\ \tilde{u}^{n} \end{Bmatrix} \tag{22}
$$

### 3.2 Dissipation and Phase Analysis

Depending on the control parameters the schemes above may lead to different dissipation and phase errors. Let the characteristic matrices of equations (18), (20) and (22) be denoted by $\mathsf{M}_c$. For $\mathsf{I}$ being an identity matrix of same rank as $\mathsf{M}_c$ and $\lambda$ an eigenvalue of $\mathsf{M}_c$, the characteristic polynomials for each scheme, given by $\det(\mathsf{M}_c - \lambda\mathsf{I})$ are,

$$
P(\lambda)_{i=\text{SE}} = -c^2\theta_i + 1 + (c^2\theta_i + c^2 - 2)\lambda + \lambda^2
$$

$$
P(\lambda)_{i=\text{SI}} = \frac{\alpha_i c^2\theta_i - \alpha_i c^2 - c^2\theta_i + c^2 + 1}{\alpha_i c^2\theta_i + 1} + \left(\frac{-2\alpha_i c^2\theta_i + \alpha_i c^2 + c^2\theta_i - 2}{\alpha_i c^2\theta_i + 1}\right)\lambda + \lambda^2
$$

$$
\begin{aligned}
P(\lambda)_{i=\text{SESM}} = {} & \beta_i\zeta_i c^2 + c^2\left(\beta_i(\gamma_i - 2\zeta_i) - \frac{\zeta_i}{2}\right)\lambda + -c^2\left((\delta_i + 3\gamma_i - 1)\beta_i + \frac{\gamma_i}{2} - \frac{3\zeta_i}{2}\right)\lambda^2 \\
& + \frac{c^2}{2}(\beta_i(6\delta_i + 6\gamma_i + 4\zeta_i - 4) + \delta_i + 4\gamma_i + \zeta_i - 1)\lambda^3 + \left(1 + \frac{c^2}{2}(\beta_i(-6\delta_i - 2\gamma_i - 2\zeta_i + 2) - 4\delta_i - 3\gamma_i - 3\zeta_i + 3)\right)\lambda^4 \\
& + \left(-2 + \frac{c^2\delta_i}{2}(2\beta_i + 3)\right)\lambda^5 + \lambda^6
\end{aligned} \tag{23}
$$

Here, the index $i$ serves to distinguish between SE, SI, and SESM schemes and their control parameters. The equations were obtained using the symbolic solver of Maple. If $\lambda$ is an eigenvalue, then $P(\lambda)_i = 0$. Given that the physical eigenvalue should closely resemble the continuous solution $\lambda = e^{ic}$, we can expand it for small $c$ as,

$$
\lambda = 1 + mc + nc^2 + qc^3 + O(c^4) \tag{24}
$$





If the schemes are to be at least second-order dissipative with respect to $c$ (see Demange et al. (2019)), $\lambda$ must also obey the relationship,

$$|\lambda| = 1 - \chi c^2 + O(c^4) \tag{25}$$

where $\chi$ is a parameter characterizing dissipation. Similarly, the phase $\tan^{-1}(\Im(\lambda_p)/\Re(\lambda_p))$ must also closely resemble the ideal phase $c$. The two conditions (24) and (25) then tie all control parameters together. From the requirement to remain formally second-order dissipative, one gets the conditions,

$$
\begin{aligned}
\text{SE}(\theta),\, & 2\chi = \theta \\
\text{SI}(\theta,\alpha),\, & 2\chi = \theta + \alpha - 1 \\
\text{SESM}(\delta,\gamma,\zeta),\, & 2\chi = \delta - \gamma - 2\zeta - \frac{1}{2}
\end{aligned}
\tag{26}
$$

with the requirement that $m = i$. Note that here $i = \sqrt{-1}$ and that each equation in (26) admits its own set of parameters, i.e., $\theta_{SI} \neq \theta_{SE}$, etc. We also reject the possibility of $m = -i$ as it immediately gives the wrong phase.

$$
\begin{aligned}
\text{SE}(\theta),\, & n = -\frac{1}{2}(1+\theta),\, q = -\frac{i}{8}(1+\theta)^2 \\
\text{SI}(\theta,\alpha),\, & n = -\frac{1}{2}(\alpha+\theta),\, q = -\frac{i}{8}(\alpha^2 + \theta^2 + 6\alpha\theta) \\
\text{SESM}(\delta,\gamma,\zeta,\beta),\, & n = -\frac{1}{2}\left(\frac{1}{2} + \delta - \gamma - 2\zeta\right),\, q = -\frac{i}{8}\left(\frac{1}{4}[\zeta(16\gamma - 16\delta + 24) + \gamma(-8\delta + 4) - 7] + 4\zeta^2 + \gamma^2 + \delta^2 + 3\delta + 4\beta\right)
\end{aligned}
\tag{27}
$$

Note that here too like (26) the parameters will be different for each scheme, i.e., $\theta_{SI} \neq \theta_{SE}$, etc. At this point, only the SE scheme is fully defined. The other schemes still have free parameters in need for optimisation - $\alpha$ for SI and $\beta, \gamma, \zeta$ for the SESM scheme. As in Shchepetkin and McWilliams (2005), $\beta$ can be set to 0.281105 for the largest stability limit. Given that dissipation is now the same (up to the second-order), one can seek to optimise for phase errors. If third-order phase accuracy is desirable, then for the SI scheme, it is only possible if $\alpha = \chi + 1/2 \pm (1/6)\sqrt{6 + 72\chi^2}$. For the SESM scheme, this gives $\gamma = -\chi^2 - 3\zeta + 1/3 - \beta$. This still leaves $\zeta$ open for optimization. It can either be obtained through further optimizing for phase accuracy or stability limit. If optimizing for stability limit, the limit can be pushed much higher like Demange et al. (2019) if one relaxes the third-order phase accuracy constraint. The results of both optimisations are as follows,

$$
\begin{aligned}
\text{With } O(c^3) \text{ phase accuracy,}\, & \zeta \approx -0.123c + 0.223 - 0.169\beta - 0.169\chi^2,\, \gamma = 1/3 - \beta - 3\zeta - \chi^2 \\
\text{Without } O(c^3) \text{ phase accuracy,}\, & \zeta \approx 0.010 - 0.135\chi,\, \gamma = 0.083 - 0.514\chi
\end{aligned}
\tag{28}
$$

Here, $\beta, \delta$ retain their earlier description. To demonstrate the benefit in terms of phase accuracy for these split-explicit schemes, we analyse their net amplitude and phase errors per baroclinic time step assuming that it consists of $M = 30$ barotropic steps in Figure 1, together with the errors of the SI scheme, against the baroclinic CFL number $c = c_p \tau k$, where $\tau$ is the baroclinic



time step. This CFL number can take high values at the largest $k \to \pi/\Delta x$, where $\Delta x$ is the mesh size. The barotropic CFL

is $M$ times smaller and stays within the stability bounds of the explicit schemes. Figure 1 shows how all tested schemes in reality are able to maintain low dissipation even for high baroclinic CFLs. The choice is then made based on phase accuracy which is very different between the implicit and explicit schemes. It is seen that both SE, and SESM schemes have orders of magnitude lower phase error compared to the SI scheme. For high $c$, the SI scheme has to be used with parameters $\alpha$ and $\theta$ ensuring strong damping of wavenumbers with large dispersive errors. Also, between the SESM and SE schemes, the SESM

seems to be the most accurate, even for same dissipation. To conclude the tests on phase accuracy, we report that irrespective of dissipation level, the split-explicit schemes will always by design provide orders of magnitude better phase accuracy compared to the semi-implicit implementation, specially in the range of high CFL numbers i.e., smaller wavelength.

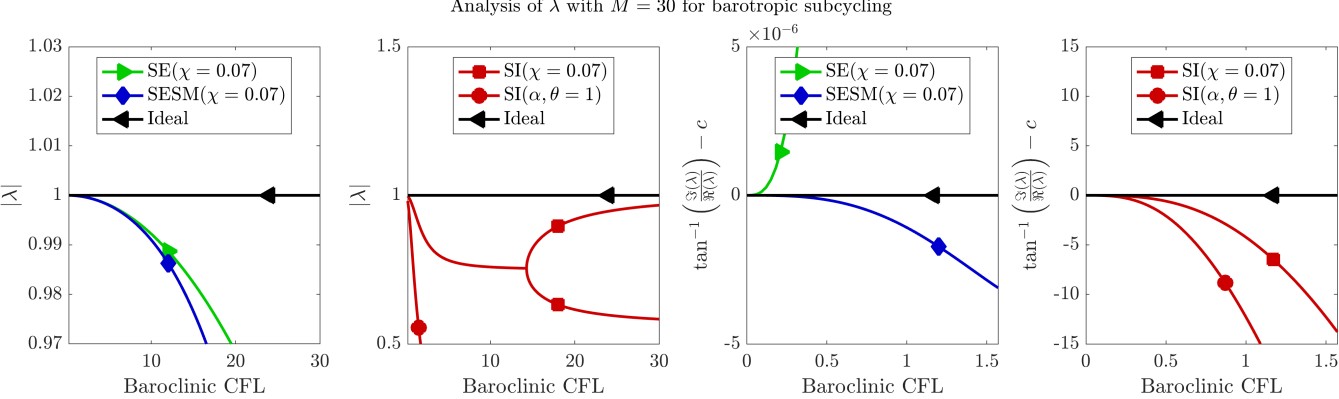

**Figure 1.** Comparison of amplitude and phase error for different schemes using same dissipation $\chi = 0.07$ and maintaining at least second order accuracy, i.e., $\zeta \approx 0.010 - 0.135\chi, \gamma = 0.083 - 0.514\chi$ for the SESM scheme as per equation 28. Additionally, the SI scheme is also plotted using its recommended configuration $\theta, \alpha = 1$ which successfully damps high phase error solutions corresponding to large CFLs.

## 4 Numerical Experiments

This section compares measurements from the new external mode solver to the existing one of FESOM2. The tests are done

for both, an idealized case, and a realisitic global setup. The idealized case is expected to highlight threshold performance of the new solver compared to the global case where its impact will also be governed by mesh non-uniformity, the presence of external forcing, complicated boundaries and bottom topography. The global case will however, crucially assess the practicality of this new solver.

### 4.1 Idealized channel

In section 3 (see Figure 1), the primary characteristics of the new schemes were already explored. In this idealized case, the solvers are tested for correct representation of the system dynamics. We use a zonally reentrant channel described in Soufflet





et al. (2016). It is 2000 km long (North-South), 500 km wide (East-West) and 4 km deep. We test 10 km meshes of different
types (triangular, quadrilateral) and unequally spaced vertical levels (40, 60). The baroclinic time step is $\tau = 720$ s, and surface
gravity wave speed $c_p = (gH_0)^{1/2} = 200$ m/s, so that a mesh cell is crossed by waves in less than 50 s. We take $M = 30$ for

the Split-Explicit solver, which means $\tau/M = 24$ s. The initial density stratification due to temperature corresponds to a zonal
jet. The zonally mean stratification and velocity are relaxed to their initial distributions. Some initial temperature perturbation
leads to an onset of baroclinic instability which is maintained through the relaxation of the zonal mean profiles. Simulations
are run for 20 years, and the last 13 years are used to compute means. Figure 2 shows that mean depth profiles for this case
are not affected by implementation of the new solver regardless of mesh configuration i.e. different mesh structure and number

of vertical layers. We can guess that this is related to the predominantly baroclinic character of the flow, so that the lower
dissipation in the new solver is not necessarily seen. The slight visible differences cannot be attributed to the new solver as
the channel undergoes large fluctuations throughout its run-time which can be verified by comparing time-evolution plots, or
temperature gradients (not shown here).

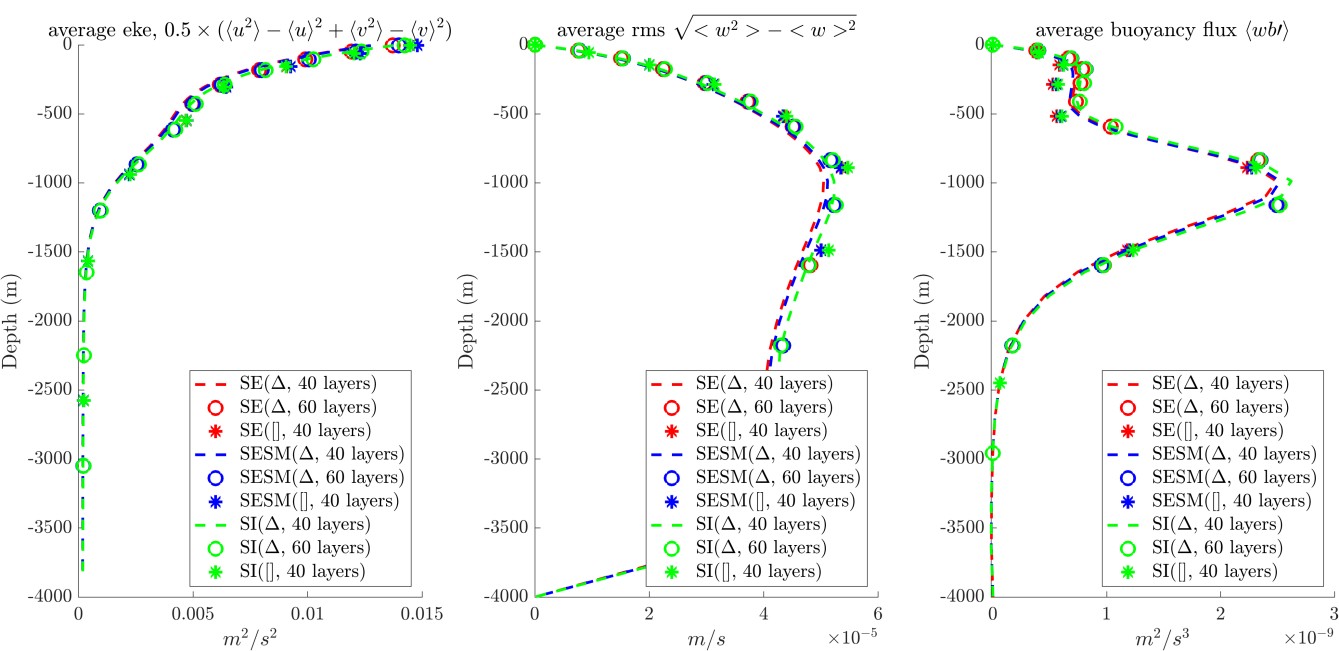

**Figure 2.** Comparison of area-averaged mean depth profiles for eddy kinetic energy ($m^2/s^2$), root mean square vertical velocity ($m/s$) and
buoyancy flux ($m^2/s^3$), with number of barotropic subcyles $M = 30$. Here $\Delta, []$ means triangular and quadrilateral mesh respectively. The
meshes used have a fixed horizontal resolution of 10 $km$ but varying vertical resolution (40 or 60 layers).





## 4.2 Realistic global ocean

For this case, we now test a more complicated case of a global ocean-sea ice simulation similar to the one used by Scholz et al. (2022). We use the standard coarse mesh of FESOM2 with a minimum resolution of 25 km north of $25^0$ N and a coarse resolution of around $1.5^0$ in the interior of the ocean, with further moderate refinements in the equatorial belt and around Antarctica. The mesh configuration consists of 47 vertical levels with a minimum layer thickness of 10 m near the surface, up to 250 m near the abyssal depth. The baroclinic time step is $\tau = 2700$ s and we take $M = 50$ for the split-explicit solver, which means that $\tau/M = 54$ s. The simulations were forced with the JRA-55do v1.4.0 reanalyses data covering the period from 1958-2019. To show the differences in the simulations carried out with the SI and the SE barotropic solvers we only show mean elevation, surface temperature and kinetic energy over the last twenty years (1999-2019) of the simulation period. Due to high similarity of SE and SESM results (as seen earlier in Figure 2 for the idealized case), only SE results are shown in Fig. 3. The differences in sea surface elevation are found to be rather small. The pattern of difference in the sea surface temperature is most likely associated with transient variability which is different in two setups. The eddy kinetic energy increases everywhere outside the equatorial belt. This increase could be associated with the reduction in overall dissipation due to use of the SE barotropic solver and the observation that the barotropic kinetic energy contributes most to the overall kinetic energy budget at mid and high latitudes, as shown in Aiki et al. (2011).

In summary, no significant difference in terms of time-averaged measurements from the new SE external mode solver was observed. For both the idealized, and the global test cases, the new SE external mode solver maintained mean dynamics close to those reported by the current SI solver.

## 5 Run-time performance and Parallel Scalability

This section compares parallel scalability of the new external mode solver to the existing one of FESOM2. Like in section 4, we again utilize the two test cases - idealized, and global, described earlier. Additionally, the two cases are also executed on different compute clusters providing for even better estimation of their general performance. Again, because of high similarities between SE and SESM parallel scalability in comparison to SI, the plots of SESM are omitted. In reality, SESM was found to be slightly less scalable than SE. The simulations are run for many model steps, and the mean total time per task the model spends for the barotropic solver is measured.

For the idealized case, the simulations were performed using the Ollie HPC of Alfred Wegener Institute equipped with Intel Xeon E5-2697 v4 (Broadwell) CPUs (308 nodes with 36 cores). To ensure sufficient amount of workload, we used a fine 2 km triangular mesh with 60 vertical layers on the same channel setup as in Soufflet et al. (2016). The mesh contains approximately $2.5 \times 10^5$ vertices, so that the setup is expected to scale almost linearly to about $10^3$ cores according to our previous experience (Koldunov et al., 2019). The baroclinic time step has been reduced to $\tau = 144$ s, and $M$ was left without changes. For this case, the simulations were run for 600 steps, i.e., 1 simulated day. As seen from figure 4 (left panel), the new external mode solver (SE) scales significantly better and is faster than the current SI solver of FESOM across all workloads. In reality, the relative speed of SE versus SI solver will depend on $M$ and on the efficiency of the preconditioner in SI, and may change.



**Figure 3.** Comparison of biases in sea surface heights ($m$), temperatures ($^0C$), diffusivities ($m^2/s$), and eddy kinetic energies ($m2/s^2$) using $M = 50$ barotropic subcyles. The SE solver uses dissipation parameter $\theta = 0.14$, and depth dependent fields are at 100 m.

Since the barotropic solver takes only a part of total time step ($10-20\%$), the improved scalability of the SE solver contributes noticeably to to the reduction of total computing time only after approximately $400$ surface vertices/core, as shown in the left panel of Figure 4. Its impact becomes significant only when parallelized beyond this limit.

For the global case, the measurements were performed using the Albedo HPC of Alfred Wegener Institute with 2xAMD Epyc-7702 CPUs (240 nodes with 128 cores). It uses the same setup and mesh from the global case of section 4. The mesh contains approximately $1.27 \times 10^5$ vertices. The baroclinic time step, and $M$ has been left unchanged to $\tau = 2700$ s, and $M = 50$ respectively. Here, simulation were run for 11680 steps, i.e., 1 simulated model year. Similar to the findings from the idealized channel case, Figure 4 (right panel) shows the new SE solver scaling similarly faster and further also for the global case. We





again observe a perceivable difference across all workloads. Also like the idealized case these performance improvements only become significant for highly parallelized workflows, i.e. less than 400 vertices/core.

In summary, the performance of the SE solver shows visible improvement in parallelization and computing time over the SI Solver across all tested workloads. The behaviour remained same over different test cases (idealized and global), and different compute resources (Ollie HPC, Albedo HPC). For less parallel workloads, the benefits are marginal but they become significant for highly parallelized workflows, i.e. approximately when vertices/core less than 400.

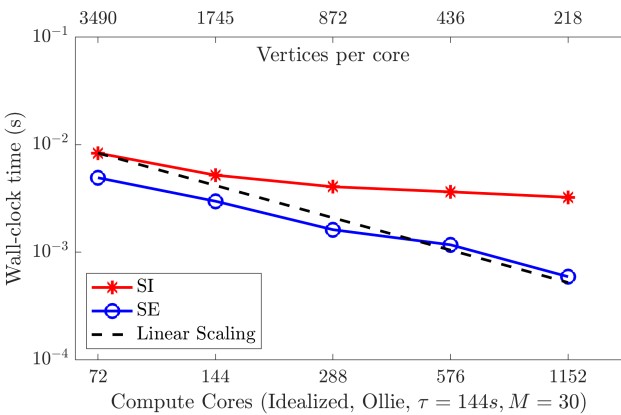 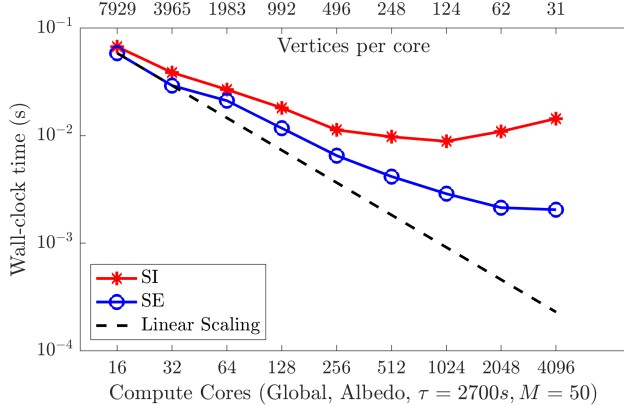

**Figure 4.** Scaling results for the idealized test case with $2\ km$ uniform mesh on Ollie HPC and the global test case with 60-25 $km$ unstructured mesh on Albedo HPC clusters. The black line indicates linear scaling and the coloured lines give the mean computing time over the parallel partitions for the solver part of the code. Here, the wall-clock time measured corresponds to a model run-time per baroclinic step.

## 6  Data and Code Availability

A preliminary implementation of the new split-explicit external mode solver within sea-ice model FESOM2 as proposed in this paper including the test cases can be found in the public repository, https://zenodo.org/doi/10.5281/zenodo.10040943.

## 7  Conclusions

The new split-explicit external mode solver proposed in this paper is more phase accurate, faster, and scalable than the SI solver used in FESOM (Danilov et al. (2017)). The dissipative asynchronous time-stepping scheme of Demange et al. (2019) is able to deliver phase accuracy orders of magnitude higher than the first order SI scheme used before. It also provides comparable phase accuracy and dissipation to the dissipatively modified AB3-AM4 scheme of Shchepetkin and McWilliams (2005) (SESM). No filtering of fast dynamics is required due to the dissipative character of SE solver. It is easier to implement compared to the SESM, and leads to very similar results as the SESM in practice. The new SE solver is a part of the adjusted time stepping of FESOM that facilitates the use of Arbitrary-Lagrangian-Eulerian vertical coordinate. As a demonstration, we extended $Z^*$ in





FESOM2 to the $\tilde{Z}$ vertical coordinate in the development version of FESOM2. The implementation of $\tilde{Z}$ is outlined in appendix B, but still needs to be tested in realistic simulations. Across different test cases using different mesh geometries and computing resources the new solver is shown to represent mean dynamics, similar to the existing solver with no significant difference. In the case of run-time performance and parallel scalability, it is shown to improve across all workloads. The improvements are shown to be specially significant for highly parallelized workloads.Kang et al. (2021) presents a semi-implicit solver for MPAS showing in contrast to the present work, that it is more computationally efficient than their split-explicit solver. While a detailed answer to the question why an opposite conclusion is reached needs a separate study, here we can only mention that by following Demange et al. (2019) we perform less external time steps per baroclinic time step than in MPAS (Ringler et al. (2013)).

We note that on unstructured meshes, a semi-implicit method can be more forgiving than a split-explicit one to the size of mesh elements. A small element on deep water will hardly affect the solution of the semi-implicit solver, but may require an increased number of barotropic substeps in a split-explicit method. This is why the semi-implicit option will be maintained in FESOM alongside the novel split-explicit option. It will however, be modified to allow more general ALE options as described in Appendix A. To conclude, this work suggest the new Split-Explicit external mode solver to be a good alternative to the existing solver of FESOM2.



## Appendix A: Adaptation of Semi-Implicit scheme in FESOM2

The main difference to the Split-Explicit method is that the elevation $\eta$ has to be defined at the same time levels as the

horizontal velocity. The elevation is therefore detached from the thicknesses, which creates some conceptual difficulty. We
consider quantities at $n-1/2$ and $n$ to be known (to start from velocity).

– Predictor step

$$\mathbf{U}^* = \mathbf{U}^{n-1/2} + \tau(\mathbf{R}_U^A + \mathbf{R}_U^C + \tilde{\mathbf{R}}_U^{PGF})^n + \tau(\mathbf{R}_U^{hV})^v - \tau g h^n \nabla_h \eta^{n-1/2}\tau$$

Here tilde implies that the contribution from the elevation to the PGF is omitted. It is taken into account explicitly (the
last term). However, since $\eta^{n+1/2}$ is unknown, we take the value from the current time level $n-1/2$. Our intention is
to get the Semi-Implicit form $\theta\eta^{n+1/2} + (1-\theta)\eta^{n-1/2}$, $1/2 \leq \theta \leq 1$ in the end. The momentum advection and Coriolis
terms are AB2 or AB3 interpolated to $n$. Implicit vertical viscosity is taken into account by solving

$$\mathbf{U}^{**} = \mathbf{U}^* + \tau(A_v \partial_z \mathbf{u}^{**})|_b^t$$

It is solved similarly as in the Split-Explicit asynchronous case.

– Corrector step

$$\mathbf{U}^{n+1/2} = \mathbf{U}^{**} - \tau\theta g h^n \nabla_h(\eta^{n+1/2} - \eta^{n-1/2})$$

This step is only written, but is evaluated after $\eta^{n+1/2}$ is available.

– The elevation step. We write

$$\eta^{n+1/2} - \eta^{n-1/2} = -\tau\nabla_h \cdot \sum_k (\alpha\mathbf{U}_k^{n+1/2} + (1-\alpha)\mathbf{U}_k^{n-1/2})$$

Here $1/2 \leq \alpha \leq 1$, which is needed for stability. This equation has to be solved together with the corrector equation. We
express $\mathbf{U}^{n+1/2}$ from the corrector equation and insert the corrector step into the elevation equation to get

$$\delta\eta = g\theta\alpha\tau^2\nabla_h \cdot H^n\nabla_h\delta\eta - \tau\nabla_h \cdot \sum_k (\alpha\mathbf{U}_k^{**} + (1-\alpha)\mathbf{U}_k^n)$$

This equation is solved for $\delta\eta = \eta^{n+1/2} - \eta^{n-1/2}$, giving $\eta^{n+1/2} = \eta^{n-1/2} + \delta\eta$.

– The corrector step is used to compute $\mathbf{U}^{n+1/2}$.

– ALE step. We write

$$h_k^{n+1} - h_k^n = -\tau[\nabla \cdot \mathbf{U}_k^{n+1/2} + w|_b^t]$$

These equations are summed vertically to give

$$H^{n+1} - H^n = -\tau\nabla_h \cdot \sum_k \mathbf{U}_k.$$

The quantity $H^{n+1} - H^0$ is the elevation at time step $n+1$. It is used to define $h^{n+1}$ for the $z^*$ vertical coordinate. The
extension to $\tilde{z}$ follows similarly to the SE case. After $h^{n+1}$ is defined, $w$ is found from the thickness equation.





- Tracers

$$(Th)_k^{n+1} - (Th)_k^n = -\tau \left[ \nabla \cdot (\mathbf{U}_k^{n+1/2} T_k^{n+1/2}) + w T_k^{n+1/2}|_b^t \right] + \tau (\nabla h^n \mathbf{K} \cdot \nabla_3 T^n)_k + \tau (K_v \partial_z T^{n+1})|_b^t$$

- By virtue of the thickness equation above,

$$\eta^{n+1/2} - \eta^{n-1/2} = \sum_k [\alpha(h_k^{n+1} - h_k^n) + (1-\alpha)(h_k^n - h_k^{n-1})]$$

(We ignore freshwater flux for simplicity, but it can be added.) The solution is

$$\eta^{n+1/2} = \sum_k (\alpha h_k^{n+1} + (1-\alpha)h_k^n) - H^0$$

If satisfied initially on cold start by formally taking $\eta^{-1/2} = 0$ and $h_k^{-1} = h_k^0$, this relationship will persist with time. However, to avoid accumulation of round-off errors, we reset $\eta^{n+1/2}$ to the right hand side of the last expression after the computations of $h_k^{n+1}$. This new $\eta^{n+1/2}$ will be used only in the next time step. The point here is that $\eta^{n+1/2}$ is computed by an iterative solver, whereby some significant digits are lost. The reset compensates for that. $\alpha = 1/2$ provides centering in time.

Both Split-Explicit and Semi-Implicit asynchronous schemes are relatively straightforward to implement. The Semi-Implicit method with $\theta = 1/2$ and $\alpha = 1/2$ is non-dissipative, and dissipation is added by shifting $\theta$ toward 1. As explained above, even though the dissipation can be well controlled by offsetting $\theta = 1/2$ only slightly, there are dispersive errors. Since the SI method is used with large Courant numbers for surface gravity waves, the contributions from such waves will come with large phase errors and should be damped. To keep centering of $\eta$, we may take $\alpha = 1/2$ and $\theta > 1/2$. FESOM in most applications
uses $\theta = 1$ and $\alpha = 1$, which implies more dissipation.

## Appendix B: Implementation of $\tilde{z}$ in FESOM2

In the case of $\tilde{z}$ vertical coordinate (Leclair and Madec (2011)) the horizontal divergence in a layer is split into fast and slow contributions. The fast one modifies layer thickness, and the slow one leads to diasurface $w$. Examples of practical implementation are provided by Petersen et al. (2015) and Megann et al. (2022). Our implementation presents a simplified version of both. The desired layer thickness is computed as,

$$h_k^{n+1} = h_k^{target} = h_k^* + h_k^{hf}$$

where $h_k^*$ corresponds to the $z^*$ coordinate, and $h_k^{hf}$ is the high-frequency component that augments $z^*$ to $\tilde{z}$. They will be defined below. The bottom depth in FESOM is cell-wise constant, whereas elevation and layer thicknesses are defined at vertices. For this reason, for a given vertex $v$, we modify thicknesses of $K' = K'(v)$ layers that do not touch topography (see Danilov et al. (2017)). The total number of layers under vertex $v$ will be denoted $K = K(v)$. We take

$$(h_k^*)^{n+1} = h_k^0(1 + \eta^{n+1}/H'), \quad H' = \sum_1^{K'} h_k^0$$





An alternative definition would be to stretch layers proportionally to their actual thickness, but Megann et al. (2022) warn that some drift in $h^*$ may present in such a case. Excluding the fixed layers, we split the divergence $D_k = \nabla \cdot (\mathbf{U}_k)$ into a quasi-barotropic part that corresponds to $h_k^*$ and the remaining quasi-baroclinic part ('quasi' because we are limited to $K'$ layers)

$$D_k = D_k^* + D_k', \quad D_k^* = h_k^0 D / H' \tag{B1}$$

where $D = \sum_{k=1}^{K} D_k$ is the vertically integrated divergence (note that all layers contribute in $D$). We will be interested in $D_k'$, which is computed as the difference between $D_k$ and $D_k^*$. We use the available thicknesses $h_k^n$ for $h_k^{n+1/2}$ in (13) to determine transports $\mathbf{U}_k^{n+1/2}$ featuring in $D_k$. After $h_k^{n+1}$ is fully specified, we re-trim $\mathbf{U}_k^{n+1/2}$ using $h^{n+1/2}$ defined as a half sum of thicknesses at full steps. Our treatment of the barotropic part is admittedly less accurate than in Petersen et al. (2015) and Megann et al. (2022), and some barotropic wave will contaminate $h_k^{hf}$. However, because of fixed bottom layers, we already introduce uncertainty from the very beginning. Since $\partial_t h_k^* = -D_k^*$, $\sum_{k=1}^{K} D_k' = 0$. The high-frequency thickness $h_k^{hf}$ will be related to $D_k'$ and should sum to zero vertically. $D'$ is split into low and high frequency parts,

$$D_k' = D_k^{lf} + D_k^{hf}$$

The low-frequency part is nudged to $D_k'$ as,

$$\partial_t D_k^{lf} = (2\pi/\tau_{lf})(D_k' - D_k^{lf})$$

where $\tau_{lf}$ is the time scale (about 5 days in Petersen et al. (2015), but larger values can be of interest according to Megann et al. (2022)). The fast frequency part is obtained by subtracting the low frequency part from $D_k'$. The high-frequency contribution to thickness is,

$$\partial_t h_k^{hf} = -D_k^{hf} - (2\pi/\tau_{hf})h^{hf} + \nabla_h(K_{hf}\nabla_h h_k^{hf}) \tag{B2}$$

The second term on the RHS damps $h_k^{hf}$ to zero over the time scale $\tau_{hf}$ (about 30 days). The last term will smooth the thickness, and the diffusivity $K_{hf}$ is determined experimentally. If $K_{hf}$ is vertically constant, $\sum_{k=1}^{K} h_k^{hf} = 0$ if it was initially so. A potential difficulty with (B2) is that $h_k^{hf}$ is not bounded. A simple procedure is implemented at present. Equation (B2) is stepped implicitly with respect to the relaxation term, and diffusion is applied in a separate step. If $(h_k^{hf})^{n+1}$ is outside bounds

for any $k$ in the column at vertex $v$, $\tau_{hf}$ is adjusted for the entire column on this time step to ensure that $(h_k^{hf})^{n+1}$ will be within the bounds, and computations of $(h_k^{hf})^{n+1}$ is repeated. While this procedure is sufficient for simple channel test case, it remains to be seen whether it will be sufficient in more realistic cases or solutions reported by Megann et al. (2022) will be needed. The field $h^{hf}$ is always damped stronger on locations close to topography to eliminate possible inconsistencies with $h^{hf} = 0$ in cells touching bottom topography. After $(h_k^{hf})^{n+1}$ is estimated, $h^{n+1}$ is available; the transports $\mathbf{U}_k^{n+1/2}$ can be

re-trimmed, and diasurface velocities can be estimated from the thickness equations.

*Author contributions.* TB, SD, KK the development of the algorithm, TB, SD, DS, PS the implementation in the prototype FESOM and main FESOM branch, all authors writing and discussions.



*Competing interests.* There are no competing interests.

*Acknowledgements.* This paper is a contribution to the projects M5 (Reducing spurious mixing and energetic inconsistencies in realistic ocean modelling applications) and S2 (Improved parameterisations and numerics in climate models) of the Collaborative Research Centre TRR 181 "Energy Transfer in Atmosphere and Ocean" funded by the Deutsche Forschungsgemeinschaft (DFG, German Research Foundation) - Projektnummer 274762653.





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
