# Peer review of "Split-explicit external mode solver in finite volume sea ice ocean model FESOM2"

_Geoscientific Model Development, 2023_

## Author Comment (AC4)

**1 Reviewer 1**

Dear Mark, We thank for the comments and took them into account. We did not have time for a rigorous convergence test proposed in the review, but will return to this question in the nearest future. Instead, we did analysis to get an estimate of the convergence order, see below. Our answers are given in blue, and the original comments in black.

Thank you, it is helpful for the ocean model development community to see your success with a split explicit method. This paper takes great care in the detailed presentation of the numerical methods for the new split explicit solver in FESOM2.

What order of convergence do you expect for your time-stepping method? Please test the convergence rate in time. The test case would need to have both slow and fast dynamics to be a useful test, but you could test them first separately.

As concerns barotropic dynamics, the inclusion of dissipation in the forward-backward method makes the barotropic time stepping first order in time. However, dissipation is small, and one expects a higher order in practice. The baroclinic time step is second order except for viscous and diffusive terms, which are first order. The convergence order of the barotropic part we found from our convergence test, is around 1.6. The test was done on the soufflet channel test case with relaxations turned off. The simulation was run for 1600s with timesteps dt = 10,8,6,2,0.5s similar to the Ange Pacifique Ishimwe paper (2023).

[Figure]

Figure 1: Order of convergence, $P \approx 1.6$

You could test JUST the barotropic part with the Kelvin Wave or Inertial-Gravity wave tests here in Bishnu et al. (2024). Just use redundant layers in the vertical. Then you can compare against an exact solution for the convergence test.

You could then use the baroclinic channel test case from Ilicak et al. (2012) and Petersen et al (2015), refine in time, and compute convergence by comparing to a short-timestep case. The duration of the simulation would need to be short (30 minutes, say) and sufficiently laminar that the results do not differ due to small differences in the turbulent flow. This test case definitely has baroclinic dynamics. It is not designed for barotropic dynamics, but should include some surface gravity waves as the SSH and barotropic eastward flow adjust to a geostrophic balance. Thanks to Ange Ishimwe (Université catholique de Louvain) for pointing out this test case during his talk at AGU Ocean Sciences. He was able to show second-order convergence for his split explicit scheme using this method in his recent paper published in December, in Ishimwe et al. (2023) figure 6. This would be a good paper to reference for comparison in the current work.

We did not do the suggested tests right now, but will try to perform them in future. As a part of revision, we carried out simple estimates of the convergence, as explained above. Also, to better illustrate the reduction in dissipation compared to the current SI scheme of FESOM, we added an additional surface gravity wave test to the paper. In this test, we measure the decay of surface gravity wave energy, which is much faster for the SI solver.

Smaller comments:

You subcycle the external mode to exactly one baroclinic time step, and do not use a filter, as explained in lines 30-34. I think this is important enough to put in the abstract, as it is a potential 2x speed-up for the barotropic stage compared to models that subcycle to n+2

Yes, added. This is indeed a very important advantage of the forward-backward dissipative time stepping proposed by Demange et al.

On equation 8, first line, I believe the sign of gH grad eta should be positive.

Yes, added

The surface flux W was dropped in equation 8. It would be good to just comment that W was added to the baroclinic (eqn 14), and not the barotropic mode, and your reasoning for that. Because you have a function at the top with the delta in equation 14, it makes sense that this is a baroclinic addition.

Made necessary changes

Fig 3 caption m2 needs superscript

Yes, added

I appreciate your description of the reasoning behind your choices. For example, the discussion of whether to add the bottom drag term to the barotropic dynamics at line 95, and the discussion on abandoning filtering at line 110. I have considered these exact issues and it is good to hear your thoughts on these.

Many thanks! Indeed, although the bottom drag and viscosity can be included in the barotropic time step, they require reconciliation with the baroclinic dynamics. Our test simulations are stable without these terms, but in realistic simulations we had to step back and add them for stability.

**2 Reviewer 2**

Dear Reviewer, many thanks for your feedback. We took them into account and made changes wherever necessary. Please find below in blue, our replies to your comments.

The authors describe the implementation of a split-explicit algorithm for the FESOM2 ocean model, using for time integration of the external mode the modified FB scheme or the modified AB3-AM4 scheme proposed by Demange et al. The implementation is described for the z-star and z-tilde vertical coordinates. Numerical integration tests are carried out for the z-star split-explicit model, and compared with the current semi-implicit algorithm of FESOM2. These tests show numerical solutions that are qualitatively close to those obtained with the semi-implicit version. They also show better scalability, particularly for highly parallelized workflows.

General comments :

The paper addresses the interesting question of how to construct a split-explicit algorithm based on the FESOM time-stepping scheme, which has the peculiarity of staggering the prognostic variables in time. Original questions arise about how to phase the time integration of 3D variables, momentum and tracers, and the time integration of the barotropic mode. However, it seems to me that the paper needs to be clarified on several points and requires major revisions.

Section 2 details the proposed split-explicit algorithm. Some clarifications seem necessary to me concerning (i) the choice made for the order of integration of the variables Ubar and eta with the modified FB scheme and the modified AB3-AM4 scheme and the implications of this choice, (ii) the initialisation of the variable Ubar for the barotropic integration, (iii) the correction of the 3d variables after completion of the barotropic integration.

We added necessary clarifications. (i) The modified FB scheme is first order with $\theta \neq 0$, and second order otherwise. However, since the dissipation is small, the order of convergence seen in our analysis is 1.6. The modified AB3-AM4 scheme should be also moved to the first order as concerns amplitudes because of the added dissipation, but remains higher-order with respect to phase errors. (ii) and (iii): We added additional schematic to the appendix for extra clarity. Briefly, Ubar is initialized by the value at the end of the previous time step (no re-initialization), and 3d velocities are made consistent with Ubar. The correction of 3d velocities is such that vertically integrated velocities agree with the barotropic velocities. The exact procedure depends on the ALE option.

Section 3 details the amplitude and phase errors of the modified FB scheme, the modified AB3-AM4 scheme and the semi-implicit FESOM scheme in the context of the SW equations. This analysis, although largely based on published results by Demange et al., could nicely illustrate the contrast between the errors produced by explicit schemes and the semi-implicit FESOM scheme. The presentation and the figure should however be improved. It seems to me that sections 2 and 3 should then be swapped, so that the SW-based results of the current section 3 introduce/motivate the implementation of the split-explicit

algorithm of current section 2.

We decided to keep the old structure, as it gives a general structure of the algorithm, which can work with different implementations of the barotropic time step. Section 3 then explains why particular selections are made.

Section 4 reports the results of numerical experiments. It is somewhat disturbing that the solutions for the idealised case show little difference between the split-explicit algorithm and the semi-implicit algorithm. This is probably due to the fact that the dynamics of this idealised case is essentially baroclinic and little affected by the representation of the external dynamics. It would have been desirable to consider an idealised case where barotropic dynamics play a more important role (for example, think of an idealised case of internal tide generation on a bathymetry by barotropic current oscillation). However, the results show a certain viability of the split-explicit algorithm and a gain in efficiency compared to the semi-implicit algorithm.

We added an additional surface gravity wave test to the paper. It illustrates more clearly that the APE decays much faster for the SI solver. We also conducted additional channel tests using a reduced bottom drag coefficient. The reduction leads to an increased barotropic component, and one starts to see that SE solver leads to a higher velocity in the deep ocean than the SI solver. We added it to the appendix Specific comments :

l37 : I suppose 'temporal interpolations' should be replaced by 'temporal integrations'?

We made necessary changes

l47 : A schematic describing the time-staggering of variables and the structure of the algorithm could be referred to here, upstream of the equations, to help the reader.

Added schematic to the appendix

l53 : Should the variable h in the expression for the pressure gradient be indexed with k?

Yes. Made necessary changes

l62 : Should the transport U be indexed with k?

Yes. Made necessary changes

l100 : It seems to me that the abbreviation SE isn't very well chosen. It would be clearer to use another which suggests that the basic time-stepping is the modified FB. For the same reason,

It is abbreviated as SE for the reason that we are proposing that as the default choice. If we go for FB, then the other one should be AB3AM4, which is a bit too long. One more option is D and SM, but then there is a dissonance with SI.

l101-102 : Here, with the SE scheme, the choice is made to integrate $U_{bar}$ before eta. With the SESM scheme, the choice will be reversed. What are the reasons for these different choices? What are the implications? In the SE case, this choice is associated with a semi-implicit discretisation of the Coriolis term, which probably requires specific numerical processing ; if so, these should be explained.On the other hand, still in the SE case, this choice leads to the expression of the mean barotropic transport (12), which would be different with

a reverse order of integration.

Added necessary clarifications to the text. In reality, everything is by historical reasons. We could start with $\eta$ in (8) and use $\theta$-representation on the gradient of elevation, to follow SM. The result would be similar. Semi-implicit Coriolis does not create problems, it is algebraic operation.

l103 : The quoted value of theta has been obtained under rather special conditions: it is the value that, for a constant stratification of N=10-2 s-1, a water column of H=4000 m and a splitting ratio of 20, allows the stabilisation of split-explicit algorithms whose stability is limited by large-scale barotropic waves. It is not clear that the algorithm proposed in this paper is constrained by these waves and that this value is relevant here. This is plausible, however, because the FESOM scheme is a variant of the FB scheme, which is used by Demange et al. in their study.

Made additional clarifications in the paper. The suggestion of Demange paper is just a parameter working in some situations. In the case of FESOM, it is a tunable parameter, but we see that the suggested value works well.

l110 : It might be clearer to change 'at the end of the barotropic step' into 'during the barotropic step'.

Agree. Made necessary changes

l114 : The reference to Shchepetkin and McWilliams on the line 112 could be moved to just after the 'traditional notation' line 114.

Made necessary changes

l120 : It is mentioned that it is not clear how to initialise Ubar (due, I suppose, to the time-staggering between U and Ubar), and it is noted that "We return to this topic below". But it's not clear to me where in the paper this is explained.

It is the paragraph starting at line 151 of the revised manuscript.

l125 : In this expression we see the correction by the mean barotropic transport $<>$ of the 3D transports $U_k^{n+1/2}$, which is then used to transport tracers. There doesn't seem to be any place in your implementation for the other transport correction that classically appears in split-explicit algorithms based on synchronous schemes, i.e the correction by the barotropic transport of the 3D transports $U_k^{n+1}$ (see for example p 384 in Shchepetkin and McWilliams). I'm a little embarrassed by the lack of an equivalent to this correction. Isn't there a cog missing to ensure consistency between the 3D and 2D integrations?

Some differences are due to asynchnonous time stepping. What is done on page 384 of Shchepetkin and McWilliams paper, is also done in our equation 12 and 13. Nothing is missing. We do not need to correct 3D $U_k^{n+1}$, as our time-stepping is staggered. We only need $U_k^{n+1/2}$ which follows the same correction as described by Shchepetkin and McWilliams on page 384. There is an issue of synchronization with the barotropic transport at $n + 1/2$, as we discuss in line 151, where we retrim $U_k^{n+1/2}$ after advecting the tracers to the barotropic velocity at $n + 1/2$. However, in practice this trimming was found to be unnecessary.

l156 : Should 'temporal interpolations' be replaced by 'temporal integrations'?

Made necessary changes

l157-159 : It seems to me that this sentence is confusing. What is done in this section is not really the analysis of the external mode in the context of split-explicit algorithms (which is done, for example, in Demange et al. section 4.2). Rather, it is the analysis of the modified FB, modified AB3-AM4 and semi-implicit FESOM schemes in the context of SW equations.

Made necessary changes for clarity. We are not analyzing the external mode. We analyze the prototype SW equations to learn about the performance of SE, SESM and SI.

l190 : The solutions of the continuous problem are $e^{+ic}$ and $e^{-ic}$.

Made additional changes for clarity. The $e^{-ic}$ solution corresponds to a wave propagating in different direction. It is a physical solution, but it will lead to the same result. We wanted to deal with one physical solution that corresponds to a wave propagating in negative direction.

l 214 : The sentence 'This CFL number ... mesh size.' is unnecessary here because the analysis is exact in space and does not depend on Delta x.

Made additional changes for clarity

fig. 1: For the two panels on the left, why not show the amplitude error of the explicit schemes over larger ranges on each of the two axes? This would show the damping for large values of c, and the loss of stability of the schemes. This would make it a little easier to compare the amplitude error of the explicit schemes with that of the semi-implicit scheme.

For the manuscript, in practice we are interested in a limited range of CFLs (for $\tau$=600 s, $c_p = 200$ m/s and maximum $k = \pi/\Delta x$ for $\Delta x = 10$ km we will have maximum CFL=20). Within this range, SE and SESM show much smaller errors than SI, which can be further reduced if M is increased. If for fixed M we go to the stability limit, SE and SESM can also show high dissipation, but we generally will avoid going there. However, for your reference, please find the expanded plots below

For the two panels on the right, the phase error could be plotted as the ratio

[Figure]

Figure 2: expanded plots

of the discrete phase velocity to the exact phase velocity. why not use the same horizontal axis as for the left panels, with c values varying between 0 and 30? This would make the figure easier to read.

We don't do till cfl 30 because it's already clear that even before that, we are approaching big errors in phase. Therefore, the solution for the semi-implicit case already needs to be highly damped. Anything beyond this wont provide us with any more meaningful information.

fig. 3 : the caption refers to diffusivities that are not traced.

Yes. Made necessary changes

---

## Author Response (AR2)

**Reviewer 2, Second review**

Dear Reviewer, thank you for your feedback. Please find below in blue, our replies to your comments.

The authors have provided some answers to the questions posed, and have made a few changes to the manuscript. I must say, however, that I am still unsatisfied with a few points that I noted in my first review and which, in my view, need to be clarified:

When I asked to clarify the choice made for the order of integration of the variables Ubar and eta with the modified FB (SE) scheme and the modified AB3-AM4 (SESM) scheme, I was not asking about the order of convergence, but about the order of integration of the two variables, i.e integrating Ubar before eta (which allows to use the newly available value for Ubar in eta integration) or conversely integrating eta before Ubar (which allows to use the newly available value for eta in Ubar integration). It has been noted that the choice of integrating Ubar before eta with the SE scheme is for historical reasons. However, it seems to me that the question of considering the inverse choice arises : this would remove the term involving $< Ubar^n + 1 >$ and $< Ubar^n >$ of the right-hand side of equation 12 : this is possibly a good thing, because as the authors note, the initialization for $< Ubar^n >$ is not well constrained (l124-126).

When we replied that the integration of $\langle\langle \overline{\mathbf{U}} \rangle\rangle$ in SE is done first for historical reason, we could have provided another answer that we just followed the original paper by Demange et al. (2019), as also mentioned in the footnote 2 of our manuscript. It is possible to do either way but that should not change the net outcome. The reviewer is right that the reverse way of integration is also possible, but it will lead to identical damping. The new equation equivalent to (12) will be simpler, but it will not lead to any practical advantages because $\langle\langle \overline{\mathbf{U}} \rangle\rangle$ is accumulated by summing the velocities appearing at the rhs of equation (8). Also the the last statement by the reviewer is a misunderstanding: we wrote that $\overline{\mathbf{U}}^n$ is not well constrained meaning that 3D transports are known at semi-integer time steps, so there is no uniquely defined way to reconcile 3d and barotropic transport. The solution we proposed is one way of reconciling and there is no issue with $\overline{\mathbf{U}}^n$. In addition, even though $\overline{\mathbf{U}}^n$ will be absent from an expression in place of (12), all velocities in that expression still depend on it. In summary, both ways of implementing SE (used by us and in Demange et al. (2019), as well as the one proposed by the reviewer) are possible and either can be followed as both will result in identical damping.

I asked to clarify the correction of the 3d variables after completion of the barotropic integration, because I had suspected that this implementation might be missing one of the two constraints which link the 2d and 3d dynamics, and which classically appear in split-explicit algorithms. As pointed out by the authors, it has been considered to correct the 3d fluxes (after having advected

tracers) to be consistent with the instantaneous barotropic flux at n+1/2, but that this correction seems unnecessary (l151-155). Why not instead correct the 3d fluxes with the half sum of $$ and $$ ?

There are three possibilities of imposing the 'second correction' in our case. The first one is just to leave the 3d transports corrected with $\langle\langle\overline{\mathbf{U}}\rangle\rangle$ (no additional correction). The second one is to retrim 3d transports with the instantaneous barotropic flux at n+1/2 (proposed by us), and the third one to use the correction proposed by the reviewer. All these corrections are similar as they are based on the estimate of the barotropic transport at n+1/2, and they all provide coupling between 2d and 3d dynamics. These possibilities differ by the amount of temporal averaging, which is strongest in the first case, and absent in the second. Indeed, the barotropic transport $\langle\langle\overline{\mathbf{U}}\rangle\rangle$ is averaged over the entire interval from $n$ to $n+1$, and the solution proposed by the reviewer is also an average over this interval. The first option and the option proposed by the reviewer are expected to provide a stronger feedback of forward-backward type, but as we already mentioned, we found that even the option proposed by us does not lead to noticeable changes compared to the first option. For this reason we did not consider the option mentioned by the reviewer. However, to demonstrate that it does not lead to noticeable changes, we added it and retested our simulations. As shown below, there is hardly any difference.

[Figure]

Figure 1: Comparison of elevation (m) distribution snapshots for the SE solver under the two different re-trimming (after 3 days) and their total available potential energy density $(m^5/s^2)$ time series. The number of barotropic subcyles $M = 30$ and $\theta = 0.14$. Here triangular mesh with sides of 10km and baroclinic time-step of $\tau = 5$min is used.

For a reminder, this test case (Section 4.2 of the manuscript) uses a simple surface gravity wave (SGW) setup where we simulate a channel (of same geometry (as Section 4.1 of the manuscript) with an initial elevation distribution which is meridionally gaussian, i.e., $\ln(\eta/A) = -(y - y_{mid})^2/\sigma^2$ where $A = 3$ meters is the amplitude and $\sigma = 200$ km is the half-bell width. The temperature

is set at $T = 20^o$c, the velocities are initialized to 0, and the simulation is run for 3 days with baroclinic time-step $\tau = 5$ mins. As seen in Figure 1 there's hardly any difference in the dissipation levels between the two re-trimming methods.

This could strengthen the coherence between the 2d and 3d dynamics, making the 3d fluxes $U_k^{n+1/2}$ 'feel' the value $< Ubar^n + 1 >$ and $< Ubar^n >$.

The 3d fluxes still 'feel' the 2d fluxes as the values of every sub-step still goes into its correction through $<< Ubar >>$. See the discussion above.

Such retrimming would parallel the treatment of layer thicknesses stated at line 135. It would also give some confidence in initializing the 2d with the final state of the previous time step, as this 2d state will have been used to correct the 3d variables. Such retrimming could be seen as the simplest analogue in this time-staggered context of the second constraint which classically appear in split-explicit algorithms.

While it is true that the reviewer's proposal would match the thickness treatment as per l135, we also state in l135 itself that this is not generally the case. While Zstar is independent of 3d flux divergences, layer algorithms like ztilde will not be. Furthermore, some layer algorithms by design would be operating with only part of the flux divergences depending on the desired objective. Therefore, in general, one will not be able to parallel the flux treatment with the thickness treatment, and neither would it be desirable.

These points are at the basis of the mechanics of the mode-splitting algorithm. I would be surprised if they did not affect the stability properties of the algorithm. It is quite possible, however, that the dissipation values used for the numerical experiments reported in the paper are so large that to counter the consequences of any inconsistencies in the algorithm. In my opinion, a more detailed study of the algorithm's stability would be desirable. However, I can imagine that this is not the purpose of the paper, which is essentially to show that 'it works'.

While exciting, exploring all types and possibilities of mode-splitting is not the objective of this paper. Indeed, our objective has always been to have an improved solver for FESOM2 that 'works'. It has been made clear from the title and throughout the manuscript that this work explores a split-explicit approach for ocean model FESOM2. The decisions made were centered around and motivated by the performance of FESOM2. And in regards to FESOM2, the new solver shows sufficient stability and abundant improvement over the previous approach across all targeted categories - dissipation, speed, and scalability.